# DIFFERENTIALLY PRIVATE GENERATIVE MODELS THROUGH OPTIMAL TRANSPORT

## ABSTRACT

Although machine learning models trained on massive data have led to breakthroughs in several areas, their deployment in privacy-sensitive domains remains limited due to restricted access to data. Generative models trained with privacy constraints on private data can sidestep this challenge and provide indirect access to the private data instead. We propose DP-Sinkhorn, a novel optimal transport-based generative method for learning data distributions from private data with differential privacy. DP-Sinkhorn relies on minimizing the Sinkhorn divergence—a computationally efficient approximation to the exact optimal transport distance—between the model and the data in a differentially private manner and also uses a novel technique for conditional generation in the Sinkhorn framework. Unlike existing approaches for training differentially private generative models, which are mostly based on generative adversarial networks, we do not rely on adversarial objectives, which are notoriously difficult to optimize, especially in the presence of noise imposed by the privacy constraints. Hence, DP-Sinkhorn is easy to train and deploy. Experimentally, despite our method's simplicity we improve upon the state-of-the-art on multiple image modeling benchmarks. We also show differentially private synthesis of informative RGB images, which has not been demonstrated before by differentially private generative models without the use of auxiliary public data.

## 1 INTRODUCTION

As the full value of data comes to fruition through a growing number of data-centric applications (e.g. recommender systems (Gomez-Uribe & Hunt, 2016), personalized medicine (Ho et al., 2020), face recognition (Wang & Deng, 2020), speech synthesis (Oord et al., 2016), etc.), the importance of privacy protection has become apparent to both the public and academia. At the same time, recent Machine Learning (ML) algorithms and applications are increasingly data hungry and the use of personal data will eventually be a necessity.

Differential Privacy (DP) is a rigorous definition of privacy that quantifies the amount of information leaked by a user participating in any data release (Dwork et al., 2006; Dwork & Roth, 2014). DP was originally designed for answering queries to statistical databases. In a typical setting, a data analyst (party wanting to use data, such as a healthcare or marketing company) sends a query to a data curator (party in charge of safekeeping the database, such as a hospital), who makes the query on the database and replies with a semi-random answer that preserves privacy. Differentially Private Stochastic Gradient Descent (DPSGD)[1] (Abadi et al., 2016) is the most popular method for training general machine learning models with DP guarantees. DPSGD involves large numbers of queries, in the form of gradient computations, to be answered quickly by the curator. This requires technology transfer of model design from analyst to curator, and strong computational capacity be present at the curator. Furthermore, if the analyst wants to train on multiple tasks, the curator must subdivide the privacy budget to spend on each task. As few institutions have simultaneous access to private data, computational resources, and expertise in machine learning, these requirements significantly limit adoption of DPSGD for learning with privacy guarantees.

To address this challenge, generative models—models with the capacity to synthesize new data—can be applied as a general medium for data-sharing (Xie et al., 2018; Augenstein et al., 2020). The

---

[1]Including any variants that use gradient perturbation for ensuring privacy.

curator first encodes private data into a generative model; then, the model is used by the analysts to synthesize similar yet different data that can train other ML applications. So long as the generative model is learned "privately", the user can protect their privacy by controlling how specific the generative model is to their own data. Differentially private learning of generative models has been studied mostly under the Generative Adversarial Networks (GAN) framework (Xie et al., 2018; Torkzadehmahani et al., 2019; Frigerio et al., 2019; Yoon et al., 2019; Chen et al., 2020). While GANs in the non-private setting have demonstrated the ability to synthesize complex data like high definition images (Brock et al., 2019; Karras et al., 2020), their application in the private setting is more challenging. This is in part because GANs suffer from training instability problems (Arjovsky & Bottou, 2017; Mescheder et al., 2018), which can be exacerbated when adding noise to the network's gradients during training, a common technique to implement DP. Because of that, GANs typically require careful hyperparameter tuning and supervision during training to avoid model collapse. This goes against the principle of privacy, where repeated interactions with data need to be avoided (Chaudhuri & Vinterbo, 2013).

Optimal Transport (OT) is another method to train generative models. In the optimal transport setting, the problem of learning a generative model is framed as minimizing the optimal transport distance, a type of Wasserstein distance, between the generator-induced distribution and the real data distribution (Bousquet et al., 2017; Peyré & Cuturi, 2019). Unfortunately, exactly computing the OT distance is generally expensive. Nevertheless, Wasserstein distance-based objectives are actually widely used to train GANs (Arjovsky et al., 2017; Gulrajani et al., 2017b). However, these approaches typically estimate a Wasserstein distance using an adversarially trained discriminator. Hence, training instabilities remain (Mescheder et al., 2018).

An alternative to adversarial-based OT estimation is provided by the Sinkhorn divergence (Genevay et al., 2016; Feydy et al., 2019; Genevay et al., 2018). The Sinkhorn divergence is an entropy-regularized version of the exact OT distance, for which the optimal transport plan can be computed efficiently via the Sinkhorn algorithm (Cuturi, 2013). In this paper, we propose DP-Sinkhorn, a novel method to train differentially private generative models using the Sinkhorn divergence as objective. Since the Sinkhorn approach does not intrinsically rely on adversarial components, it avoids any potential training instabilities and removes the need for early stopping. This makes our method easy to train and deploy in practice. As a side, we also develop a simple yet effective way to perform conditional generation in the Sinkhorn framework, by forcing the optimal transport plan to couple same-label data closer together. To the best of our knowledge, DP-Sinkhorn is the first fully OT-based approach for differentially private generative modeling.

Experimentally, despite its simplicity DP-Sinkhorn achieves state-of-the-art results on image-based classification benchmarks that use data generated under differential privacy for training. We can also generate informative RGB images, which, to the best of our knowledge, has not been demonstrated by any generative models trained with differential privacy and without auxiliary public data.

We make the following contributions: (i) We propose DP-Sinkhorn, a flexible and robust optimal transport-based framework for training differentially private generative models. (ii) We introduce a simple technique to perform label-conditional synthesis in the Sinkhorn framework. (iii) We achieve state-of-the-art performance on widely used image modeling benchmarks. (iv) We present informative RGB images generated under strict differential privacy without the use of public data.

## 2 BACKGROUND

### 2.1 NOTATIONS AND SETTING

Let $\mathcal{X}$ denote a sample space, $\mathcal{P}(\mathcal{X})$ all possible measures on $\mathcal{X}$, and $\mathcal{Z} \subseteq \mathbb{R}^d$ the latent space. We are interested in training a generative model $g : \mathcal{Z} \mapsto \mathcal{X}$ such that its induced distribution $\mu = g \circ \xi$ with noise source $\xi \in \mathcal{P}(\mathcal{Z})$ is similar to observed $\nu$ through an independently sampled finite sized set of observations $D = \{y\}^N$. In our case, $g$ is a trainable parametric function with parameters $\theta$.

### 2.2 GENERATIVE LEARNING WITH OPTIMAL TRANSPORT

Optimal Transport-based generative learning considers minimizing variants of the Wasserstein distance between real and generated distributions (Bousquet et al., 2017; Peyré & Cuturi, 2019).

Two key advantages of the Wasserstein distance over standard GANs, which optimize the Jensen-Shannon divergence (Goodfellow et al., 2014), are its definiteness on distributions with non-overlapping supports, and its weak metrization of probability spaces (Arjovsky et al., 2017). This prevents collapse during training caused by discriminators that are overfit to the training examples.

Methods implementing the OT framework use either the primal or the dual formulation. For generative learning, the dual formulation has been more popular. Under the dual formulation, the distance between the generator-induced and the data distribution is computed as the expectation of potential functions over the sample space. In WGAN and variants (Arjovsky et al., 2017; Gulrajani et al., 2017a; Miyato et al., 2018), the dual potential is approximated by an adversarially trained discriminator network. While theoretically sound, these methods still encounter instabilities during training since the non-optimality of the discriminator can produce arbitrarily large biases in the generator gradient (Bousquet et al., 2017). The primal formulation involves solving for the optimal transport plan—a joint distribution over the real and generated sample spaces. The distance between the two distributions is then measured as the expectation of a point-wise cost function between pairs of samples as distributed according to the transport plan. Thus, the properties of the point-wise cost function are of great importance. In practice, sufficiently convex cost functions allow for an efficient optimization of the generator. It is also possible to learn cost functions adversarially (Szabó & Sriperumbudur, 2017). However, as discussed earlier, adversarial training often comes with additional challenges, which can be especially problematic in the differentially private setting.

In general, finding the optimal transport plan is a difficult optimization problem due to the constraints of equality between its marginals and the real and generated distributions. Entropy Regularized Wasserstein Distance (ERWD) imposes a strongly convex regularization term on the Wasserstein distance, making the OT problem between finite samples solvable in linear time (Peyré et al., 2019). Given a positive cost function $c : \mathcal{X} \times \mathcal{X} \mapsto \mathbb{R}^+$ and $\epsilon \geq 0$, the EERWD is defined as:

$$W_{c,\epsilon}(\mu,\nu) = \min_{\pi \in \Pi} \int c(x,y)\pi(x,y) + \epsilon \int \log \frac{\pi(x,y)}{d\mu(x)d\nu(y)} d\pi(x,y) \tag{1}$$

where $\Pi = \left\{ \pi(x,y) \in \mathcal{P}(\mathcal{X} \times \mathcal{X}) | \int \pi(x,\cdot)dx = \nu, \int \pi(\cdot,y)dy = \mu \right\}$. Sinkhorn divergence uses cross correlation terms to cancel out the entropic bias introduced by ERWD. This results in faithful matching between the generator and real distributions. In this paper, we use the Sinkhorn divergence as defined in Feydy et al. (2019).

**Definition 2.1** *(Sinkhorn Divergence) The Sinkhorn divergence between measures $\mu$ and $\nu$ is defined as:*

$$S_{c,\epsilon}(\mu,\nu) = 2W_{c,\epsilon}(\mu,\nu) - W_{c,\epsilon}(\mu,\mu) - W_{c,\epsilon}(\nu,\nu) \tag{2}$$

Works on the efficient computation of the Sinkhorn divergence have yielded algorithms that converge to the optimal transport plan within tens of iterations (Cuturi, 2013; Feydy et al., 2019). Gradient computation follows by taking the Jacobian vector product between the cost matrix Jacobian and the transport weights, which is implemented in many auto-differentiation frameworks.

When compared to WGAN, learning with the Sinkhorn divergence has distinct differences. First, the Sinkhorn divergence is computed under the primal formulation of OT, whereas WGAN's loss is computed under the dual formulation. While both are approximations to the exact Wasserstein distance, the source of the approximation error differs. The Sinkhorn divergence uses entropic regularization to ensure linear convergence when finding the optimal transport plan $\pi$ (Cuturi, 2013). Its two sources of error are the suboptimality of the transport plan and bias introduced by entropic regularization. With the guarantee of linear convergence, $\pi$ can converge to optimality by using enough iterations, thereby allowing control of the first error. The second source of error can be controlled by using small values of $\epsilon$, which we found to work well in practice. In contrast, WGAN's source of error lies in the sub-optimality of the dual potential function. Since this potential function is parameterized by an adversarially trained deep neural network, it enjoys neither convergence guarantees nor feasibility guarantees. Furthermore, the adversarial training scheme can produce oscillatory behavior, where the discriminator and generator change abruptly every iteration to counter the strategy of the other player from the previous iteration (Mescheder et al., 2017). These shortcomings contribute to WGAN's problems of non-convergence, which in turn can lead to mode dropping. In contrast, training with the Sinkhorn divergence does not involve any adversarial training at all, converges more stably, and reaps the benefits of OT metrics at covering modes.

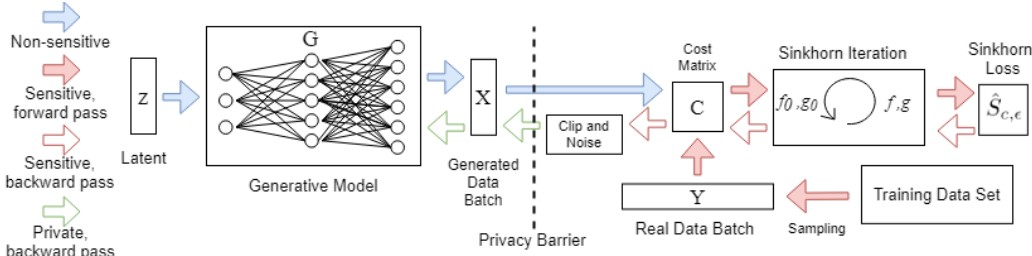

Figure 1: Flow diagram of DP-Sinkhorn for a single training iteration: Sensitive training data is combined with non-sensitive generated data in the cost matrix. Then, the loss is calculated using the Sinkhorn algorithm. In the backward pass, we impose a privacy barrier behind the generator by clipping and adding noise to the gradients at the generated image level, similar to Chen et al. (2020).

## 2.3 DIFFERENTIAL PRIVACY

The current gold standard for measuring the privacy risk of data releasing programs is the notion of differential privacy (DP) (Dwork et al., 2006). Informally, DP measures to what degree a program's output can deviate between adjacent input datasets—sets which differ by one entry. For a user contributing their data, this translates to a guarantee on how much an adversary could learn about them from observing the program's output. In this paper, we are interested in the domain of image-label datasets where each image and its semantic label constitute an entry.

**Gradient perturbation** with the Gaussian mechanism is the most popular method for DP learning of parametric models. During stochastic gradient descent (SGD) and variants, the parameter gradients are clipped in 2-norm by a constant, and then Gaussian noise is added. By adding a sufficient amount of noise, the gradients can satisfy DP requirements. Then, the *post-processing* property of differential privacy (Dwork & Roth, 2014) guarantees that the parameters are also private. As SGD involves computing the gradients on randomly drawn batches of data for multiple iterations, two other properties of DP, *composition* and *subsampling* are required to analyze the privacy of the algorithm. Rényi Differential Privacy (RDP) is a well-studied formulation of privacy that allows tight composition of multiple queries, and can be easily converted to standard definitions of DP.

**Definition 2.2** *(Rényi Differential Privacy) A randomized mechanism $\mathcal{M} : \mathcal{D} \rightarrow \mathcal{R}$ with domain $\mathcal{D}$ and range $\mathcal{R}$ satisfies $(\alpha, \epsilon)$-RDP if for any adjacent $d, d' \in \mathcal{D}$ it holds that*

$$D_\alpha(\mathcal{M}(d)|\mathcal{M}(d')) \leq \epsilon, \tag{3}$$

*where $D_\alpha$ is the Rényi divergence of order $\alpha$. Also, any $\mathcal{M}$ that satisfies $(\alpha, \epsilon)$-RDP also satisfies $(\epsilon + \frac{\log 1/\delta}{\alpha-1}, \delta)$-DP (Mironov, 2017).*

For clipping threshold $\Delta$ and standard deviation of Gaussian noise $\sigma$, the Gaussian mechanism satisfies $(\alpha, \alpha\Delta^2/(2\sigma^2))$-RDP (Mironov, 2017). Subsampling the dataset into batches also improves privacy. The effect of subsampling on the Gaussian mechanism under RDP has been studied in (Wang et al., 2019; Balle et al., 2018; Zhu & Wang, 2019). Privacy analysis of a gradient-based learning algorithm entails accounting for the privacy cost of single queries (possibly with subsampling), summing up the privacy cost across all queries (i.e. training iterations in our case), and then choosing the best $\alpha$. A more thorough discussion of DP can be found in Appendix B.

## 3 DIFFERENTIALLY PRIVATE SINKHORN WITH CLASS CONDITIONING

We propose DP-Sinkhorn (Algorithm 1), an OT-based method to learn differentially private generative models, avoiding the training instability problems characteristic for GAN-based techniques. We use the empirical Sinkhorn loss between batches of real and generated data as objective and define the cost function as a simple pixel-wise L2-loss. Class conditioning is achieved by biasing the optimal transport plan to couple images of the same class together. Privacy protection is enforced by clipping and adding noise to the gradients of the generated images during backpropagation.

**Empirical Sinkhorn Divergence**   Definition 2.1 requires integration over the sample space. In practice, we use an empirical Sinkhorn divergence computed on batches of random samples.

**Definition 3.1** *(Empirical Sinkhorn loss) The empirical Sinkhorn loss computed over a batch of $N$ generated examples and $M$ real examples is defined as:*

$$\hat{S}_{c,\epsilon}(\mathbf{X}, \mathbf{Y}) = 2C_{XY} \odot P^*_{\epsilon,X,Y} - C_{XX} \odot P^*_{\epsilon,X,X} - C_{YY} \odot P^*_{\epsilon,Y,Y} \qquad (4)$$

*where $\mathbf{X}$ and $\mathbf{Y}$ are uniformly sampled batches from the generator and real distribution respectively. For two samples $A \in \mathcal{X}^N$ and $B \in \mathcal{X}^M$, $C_{AB}$ is the cost matrix between $A$ and $B$, and $P^*_{\epsilon,A,B}$ is an approximate optimal transport plan that minimizes Eqn. 1 computed over $A$ and $B$.*

Algorithm 1 describes how Equation 4 is computed, while additionally modifying the gradient by adding noise and clipping to implement the chosen mechanism for differential privacy (see below). Please also see Appendix A for more details.

**Class Conditioning**   To conditionally generate images given a target class, we inject class information to both the generator and the Sinkhorn loss during training. For the generator, we supply the class label either by concatenating an embedding of it with the sampled latent code (on simple DCGAN-style generators), or by using class conditioned batch-norm layers (on BigGAN-style generators). For the loss function, we concatenate a scaled one-hot class embedding to both the generated images and real images. Intuitively, this works by increasing the cost between image pairs of different classes, hence shifting the weight of the transport plan ($P^*_\epsilon$ in Eq. 4) towards class-matched pairs. The scaling constant $\alpha_c$ determines the importance of class similarity relative to image similarity. Uniformly sampled labels are used for the generated images and the real labels are used for real images. Let $l_x$ and $l_y$ denote the labels of x and y, and $[a, b]$ denote concatenation of a and b. The class-conditional pixel-wise and label loss is:

$$c_{cond}([\mathrm{x}, l_x], [\mathrm{y}, l_y]) = \left|\left|[\mathrm{x}, \alpha_c * \mathrm{onehot}(l_x)]^T - [\mathrm{y}, \alpha_c * \mathrm{onehot}(l_y)]^T\right|\right|_2^2. \qquad (5)$$

When $c_{cond}$ is used for computing $\hat{S}_{c,\epsilon}$, the resulting optimal transport distance is the cost of transporting the *joint* distribution of generated images and labels to the real distribution. Compared to generating without class conditioning, this means choosing $\mathcal{X}$ to denote the joint space of images and labels instead of the image space. All other formulation remains unchanged. Furthermore, since $||[\mathrm{a}, \mathrm{b}]||_2^2 = ||\mathrm{a}||_2^2 + ||\mathrm{b}||_2^2$, we can also interpret $c_{cond}$ as separately calculating the cost of transporting images and labels, and then performing a weighted sum. Despite computing the cost separately, our method does not assume that images and labels are independent. Instead, in our formulation the joint distribution $p(\mathrm{image, label})$ is defined through a decomposition as $p(\mathrm{image|label})p(\mathrm{label})$ and novel image-label pairs are synthesized by ancestral sampling. It is worth mentioning that a squared L2 norm as label cost may seem unintuitive. However, we found this to work very well and we did not observe any instances of class-conditioning failure. Hence, we chose to stick to the simplest choice, in particular because we are using the L2 norm already for the image cost itself. Future work may consider tailored cost functions for more complex label spaces.

Note that, to the best of our knowledge, it has not been shown before how to perform class-conditional generation in the Sinkhorn divergence generative framework. Our simple yet efficient trick can similarly be used in non-private learning settings.

**Privacy Protection**   Rather than adding noise to the gradients of the generator parameters $\theta$, we add noise to the gradients of the generated images $\nabla_{\mathbf{X}}\hat{S}$, which is then backpropagated to $\theta$. This is private because the input to generators (the latent code) is randomly drawn and independent from data, leaving the only connection to data at the output. Compared to $\nabla_\theta \hat{S}$, the dimension of $\nabla_{\mathbf{X}}\hat{S}$ is independent from the network architecture. This allows us to train larger networks without requiring more aggressive clipping, as $||\nabla_\theta \hat{S}||_2$ scales as $\sqrt{dim(\theta)}$. This method has been independently proposed by Chen et al. (2020). In each iteration, gradient descent updates the generator parameters by backpropagating the "image gradient" $\nabla_{\mathbf{X}}\hat{S}$. We clip this term such that $||\nabla_{\mathbf{X}}\hat{S}||_2 \leq C$. The sensitivity is thus $\max_{\mathbf{Y},\mathbf{Y}'} ||\nabla_{\mathbf{X}}\hat{S}(\mathbf{X}, \mathbf{Y}) - \nabla_{\mathbf{X}}\hat{S}(\mathbf{X}, \mathbf{Y}')||_2 \leq 2C$. By adding Gaussian noise with scale $2C\sigma$, the mechanism satisfies $(\alpha, \frac{\alpha C^2}{2\sigma^2})$-RDP (Mironov, 2017). We use the RDP accountant with Poisson subsampling proposed in Zhu & Wang (2019) for privacy composition. Note that the batch size of $\mathbf{X}$ is kept fixed, while batch size of $\mathbf{Y}$ follows a binomial distribution due to Poisson subsampling.

## 4 EXPERIMENTS

We conduct experiments on conditional image synthesis tasks since our focus is on the generation of high-dimensional data with privacy protection. We evaluate our method with respect to both visual quality and data utility for downstream learning tasks. To compare with previous work, all models are trained under a privacy budget of $\varepsilon = 10$.

### 4.1 EXPERIMENTAL SETUP

**Datasets** We consider 3 image datasets: MNIST (LeCun et al., 1998), Fashion-MNIST (Xiao et al., 2017), and CelebA (Liu et al., 2015) downsampled to 32x32 resolution. For MNIST and Fashion-MNIST, generation is conditioned on the regular class labels while on CelebA we condition on gender.

**Metrics** In all experiments, we compute metrics against a synthetic dataset of 60k image-label pairs sampled from the model. For a quantitative measure of visual quality, we report FID (Heusel et al., 2017). We compute FID scores between our synthetic datasets of size 60k and the full test data (either 10k or 19962 images). To measure the utility of gen-

---

**Algorithm 1** DP-Sinkhorn

$L$ is number of categories, $\mathcal{X}$ is sample space. $M$ is size of private data set. $backprop$ is a reverse mode auto-differentiation function that takes 'out', 'in' and 'grad weights' as input and computes the Jacobian vector product $J_{\text{in}}(\text{out}) \cdot$ grad weights. Poisson Sample and $\hat{W}$ are defined in Appendix C.

---

**Input:** private data set $d = \{(\mathrm{y},\mathrm{l}) \in \mathcal{X} \times \{0,...,L\}\}^M$, sampling ratio $q$, noise scale $\sigma$, clipping coefficient $\Delta$, generator $g_\theta$, learning rate $\alpha$, entropy regularization $\epsilon$, total steps $T$.
**Output:** $\theta$
$n = q * M$
**for** $t = 1$ **to** $T$ **do**
    Sample $\mathbf{Y} \leftarrow$ Poisson Sample$(d, q)$,
    $Z \leftarrow (\mathrm{z}_i)_{i=1}^n \overset{i.i.d.}{\sim} \text{Unif}(0,1)$
    $L_x \leftarrow \{\mathrm{l}_i\}_{i=1}^n \overset{i.i.d.}{\sim} \text{Unif}(0,...,L)$
    $\mathbf{X} \leftarrow \{x_i = g_\theta(\mathrm{z}_i, \mathrm{l}_i)\}_{i=1}^n$
    $\text{grad}_{\mathbf{X}} \leftarrow \nabla_{\mathbf{X}} \left[ 2\hat{W}_\epsilon(\mathbf{X}, \mathbf{Y}) - \hat{W}_\epsilon(\mathbf{X}, \mathbf{X}) \right]$
    $\text{grad}_{\mathbf{X}} \leftarrow clip(\text{grad}_{\mathbf{X}}, \Delta) + 2\Delta\sigma\mathcal{N}(\vec{0}, \mathbb{I})$
    $\text{grad}_\theta \leftarrow backprop(\mathbf{X}, \theta, \text{grad}_{\mathbf{X}})$
    $\theta \leftarrow \theta - \alpha * Adam(\text{grad}_\theta)$
**end for**

---

erated data, we assess the class prediction accuracy of classifiers trained with synthetic data on the real test sets. We consider logistic regression, MLP, and CNN classifiers. Previous work also reports classification accuracies on a large suite of classifiers from scikit-learn. We omit them, since we focus on images which are best processed via neural network-based classifiers.

**Architectures & Hyperparameters** For MNIST and Fashion-MNIST experiments, we adopt the generator architecture from DCGAN (Radford et al., 2015). For CelebA experiments, we adopt the CIFAR-10 generator configuration from BigGAN (Brock et al., 2019). Our preliminary experiments found that using the deeper BigGAN architecture did not provide significant improvements on MNIST and Fashion-MNIST datasets, hence we only report results with the DCGAN-based generator for these two datasets. Additional details can be found in Appendix D.3.

**Privacy Implementation** All our models are implemented in PyTorch. We implement the gradient sanitization mechanism by registering a backward hook to generator output, and we do not see a significant impact on runtime. MNIST and Fashion-MNIST experiments target $(10, 10^{-5})$-DP while CelebA experiments target $(10, 10^{-6})$-DP. Details are in Appendix D.3.

### 4.2 EXPERIMENTAL RESULTS ON STANDARD BENCHMARKS

In Table 1, we compare the performance of DP-Sinkhorn with other methods on MNIST and Fashion-MNIST. Given the same privacy budget, DP-Sinkhorn generates more informative examples than previous methods, as demonstrated by the higher accuracy achieved by the downstream classifier. In particular, on the more visually complex Fashion-MNIST, DP-Sinkhorn's lead is especially pronounced, beating previous state-of-the-art results by a significant margin.

The FID of images generated by DP-Sinkhorn is lower than all baselines, except GS-WGAN (Chen et al., 2020), likely because of the slight blur in images generated by DP-Sinkhorn, as shown in Figure 2. We hypothesize that this is due to the simple $L2$ cost and that improved cost functions based on other kernels may perform better. However, in this work our focus is to keep the model simple and focus on downstream applications, for which our model works best. It is also worth noting that GS-WGAN uses a more complex adversarial setup and requires an auxiliary discriminator

network, while we rely on the robust Sinkhorn framework. Furthermore, FID scores are based on an ImageNet-trained Inception network and are not well suited for analyzing small grayscale images.

**Ablations**  We study the effect of image gradient noise (perturbing $\nabla_{\mathbf{X}}\hat{S}$) versus parameter gradient noise (perturbing $\nabla_\theta\hat{S}$). Performance of the parameter gradient noise variant of DP-Sinkhorn is shown in Table 1 in the row DP-Sinkhorn ($\nabla_\theta\hat{S}$). We tuned the clipping bound $\Delta$ separately for this variant, while other hyper-parameters were kept fixed. We see that DP-Sinkhorn ($\nabla_\theta\hat{S}$) still outperforms GS-WGAN in downstream classification accuracy, and is only slightly behind DP-Sinkhorn in most measures, except for FID score on MNIST. Further inspection (in Appendix E) shows that images generated by DP-Sinkhorn ($\nabla_\theta\hat{S}$) are well formed but have noisy edges. Since real MNIST images have sharper lines than Fashion-MNIST, FID is impacted more severely by this noise. The observation that DP-Sinkhorn outperforms GS-WGAN and the other baselines in the classification task, despite resulting in slightly noisy images compared to GS-WGAN, suggests that our samples are more diverse, which leads to better generalization when training the downstream classifier. We hypothesize that the baselines suffer from some amount of mode dropping compared to our method. We attribute this to our robust optimal transport-based training approach.

**Robustness**  In Tables 4 and 5 in Appendix E we report additional results on training DP-Sinkhorn with a variety of different hyperparameters (optimizer, learning rate, batch size). The majority of these experiments train in a stable manner, reaching competitive or state-of-the-art performance. This suggests that our method is indeed robust against the choice of hyperparameters.

### 4.3 EXPERIMENTAL RESULTS ON CELEBA

We also evaluate DP-Sinkhorn on downsampled CelebA. To the extent of our knowledge, no DP generative learning method has been applied on such RGB image data without accessing public data. We evaluate whether DP-Sinkhorn is able to synthesize RGB images that are informative for downstream classification. Furthermore, we study whether an adversarial learning scheme is helpful for learning this dataset. We test this by using an adversarially trained feature extractor in the cost function. Given feature extractor $\phi$, we modify the cost function as:

$$c_{adv}([\mathrm{x}, l_x], [\mathrm{y}, l_y]) = \left|\left|[\mathrm{x}, \alpha_c * \mathrm{onehot}(l_x), \frac{\alpha_a \phi(\mathrm{x})}{||\phi(\mathrm{x})||_2}]^T - [\mathrm{y}, \alpha_c * \mathrm{onehot}(l_y), \frac{\alpha_a \phi(\mathrm{y})}{||\phi(\mathrm{y})||_2}]^T\right|\right|_2^2.$$
(6)

The output of the feature extractor is normalized and then concatenated to the image and class label. Training proceeds through alternating updates. To make this learning scheme private, we randomly split the training dataset into $k$ partitions and train one discriminator per partition. This way, the gradient (with respect to generated image) computed on each partition is only dependent on that partition, hence still benefiting from privacy amplification by subsampling. Privacy accounting for this setting is performed through Wang et al. (2019), which analyzes fixed-size batch subsampling.

We find that while using the adversarial feature extractor is beneficial in the non-private case, it did not provide significant improvements in the private case, as shown in Table 2. In theory, using learned feature extractors can provide a sharper loss landscape which should be beneficial to learning (Li et al., 2017). We hypothesize that clipping and gradient noise counteracts the effect of gradient shaping provided by the adversarial feature extractor, resulting in similar learning performance as in the non-adversarial approach. Despite its simplicity when using only an $L2$ cost function, DP-Sinkhorn generates informative images for gender classification, (uninformative images would correspond to a $\approx 50\%$ classification ratio). Qualitatively, Figure 3 shows that DP-Sinkhorn can learn rough representations of each semantic class (male and female) and produces some in-class variations. Visual fidelity of the generated images may be improved in the future through fine-tuning of the adversarial scheme, use of tailored kernels as cost function, or tighter privacy accounting.

## 5 RELATED WORKS

The task of learning generative models on private data has been tackled by many prior works. The general approach is to introduce privacy-calibrated noise into the model parameter gradients during training. While various GAN-based approaches have been introduced (Xie et al., 2018; Torkzadehmahani et al., 2019; Frigerio et al., 2019; Yoon et al., 2019; Chen et al., 2020), it is well

Table 1: Comparison of DP image generation results on MNIST and Fashion-MNIST at $(\epsilon, \delta) = (10, 10^{-5})$-DP. Results for other methods (G-PATE (Long et al., 2019), DP-MERF & DP-MERF AE (Harder et al., 2020), DP-CGAN (Torkzadehmahani et al., 2019), GS-WGAN (Chen et al., 2020)) are from Chen et al. (2020). Results averaged over 5 runs of synthetic dataset generation.

| Method | DP-$\epsilon$ | MNIST | | | | Fashion-MNIST | | | |
| | | FID | Acc (%) | | | FID | Acc (%) | | |
| | | | Log Reg | MLP | CNN | | Log Reg | MLP | CNN |
| Real data | $\infty$ | 1.6 | 92.2 | 97.5 | 99.3 | 2.5 | 84.5 | 88.2 | 90.8 |
| Non-priv Sinkhorn | $\infty$ | 84.1 | 88.6 | 88.2 | 87.9 | 105.2 | 77.6 | 78.7 | 72.8 |
| G-PATE | 10 | 177.2 | 26 | 25 | 51/80.9[1] | 205.8 | 42 | 30 | 50/69.3[1] |
| DP-CGAN | 10 | 179.2 | 60 | 60 | 63 | 243.8 | 51 | 50 | 46 |
| DP-MERF[2] | 10 | 247.5 | 66 | 63 | 63 | 267.8 | 59 | 56 | 64 |
| DP-MERF AE[2] | 10 | 161.1 | 54 | 55 | 68 | 213.6 | 50 | 56 | 62 |
| GS-WGAN | 10 | **61.3** | 79 | 79 | 80 | **131.3** | 68 | 65 | 65 |
| DP-Sinkhorn ($\nabla_\theta \hat{S}$) | 10 | 218.6 | 79.7 | 79.9 | 80.7 | 213.9 | 70.5 | 71.2 | 68.6 |
| DP-Sinkhorn | 10 | 124.3 | **82.0** | **80.8** | 79.9 | 193.8 | **73.4** | **72.2** | 67.5 |

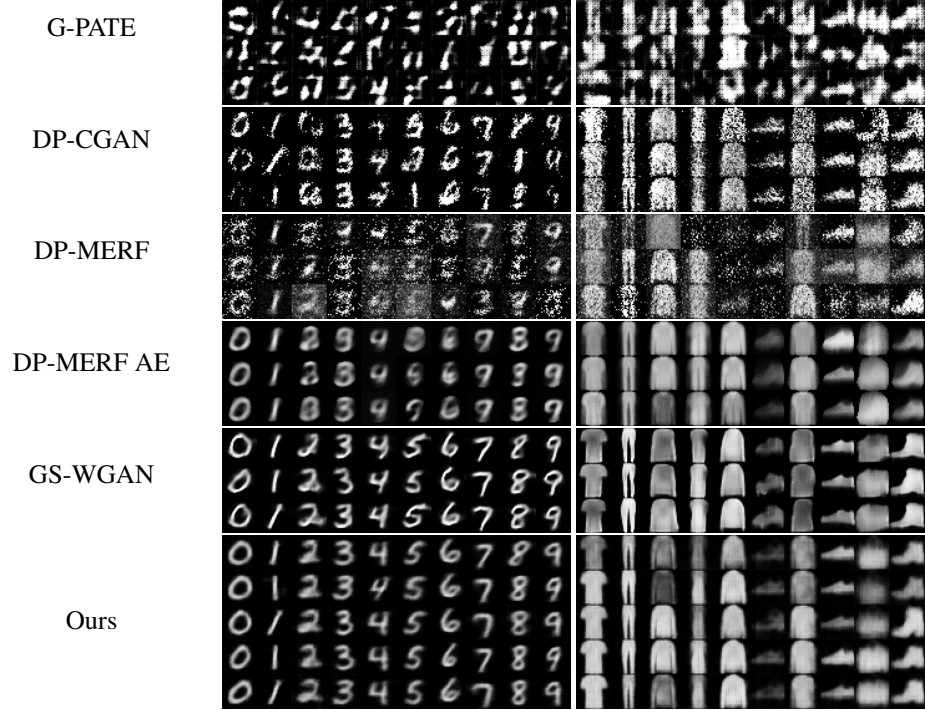

Figure 2: Comparisons on MNIST and Fashion-MNIST between methods for private image generation at $(10, 10^{-5})$-DP. The first 5 rows showing other methods are taken from Chen et al. (2020).

documented that GANs are unstable during training (Arjovsky & Bottou, 2017; Mescheder et al., 2018). As discussed, this is critical in the context of DP, where the imposed gradient noise can increase training instabilities and where interaction with private data should be limited. It is worth noting that Chen et al. (2020) proposed the same privacy barrier mechanism like us by applying gradient noise on the generated images rather than the generator parameters.

Other generative models have also been studied in under DP setting. Acs et al. (2018) partitions the private data in clusters and learns separate likelihood-based models for each cluster. Harder et al. (2020) uses MMD with random Fourier features. While these works do not face the same stability issues as GANs, their restricted modelling capacity results in these methods mostly learning

---

[1]The G-PATE authors report much more accurate classification results than reported in Chen et al. (2020). The visual quality of samples in both papers is roughly the same.

[2]DP-MERF is designed for the low-$\varepsilon$ regime and does not make use of the extra privacy budget from $\varepsilon = 10$.

Table 2: Differentially private image generation results on downsampled CelebA .

| Method | $(\epsilon, 10^{-6})$-DP | FID | Acc (%) | |
| --- | --- | --- | --- | --- |
| | | | MLP | CNN |
| Real data | $\infty$ | 1.1 | 91.9 | 95.0 |
| Adversarial Sinkhorn (non-priv) | $\infty$ | 72.8 | 78.1 | 78.2 |
| Pixel Sinkhorn (non-priv) | $\infty$ | 140.7 | 78.9 | 77.9 |
| Adversarial DP-Sinkhorn | 10 | 187.0 | 74.7 | 74.5 |
| Pixel DP-Sinkhorn | 10 | 168.4 | 76.2 | 75.8 |

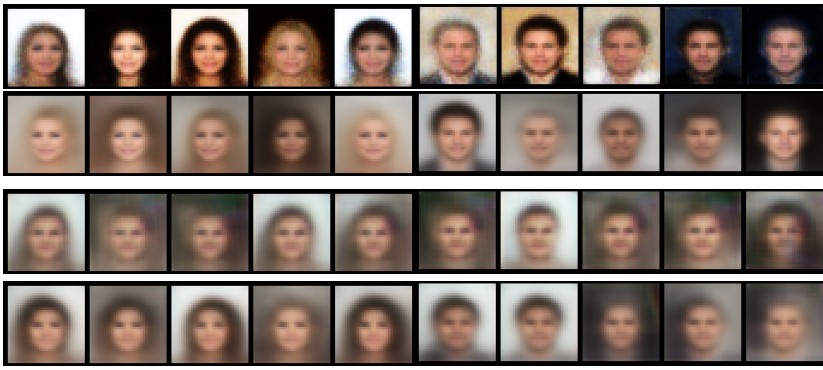

Figure 3: Images Generated on CelebA dataset. From top to botton: Adversarial Sinkhorn, Pixel Sinkhorn, Adversarial DP-Sinkhorn, Pixel DP-Sinkhorn.

prototypes for each class. Lastly, while Takagi et al. (2020) produced strong empirical results, their privacy analysis relies on the use of Wishart noise on sample covariance matrices, which has been proven to leak privacy (Sarwate, 2017). Hence, their privacy protection is invalid in its current form.

There are various works on incorporating non-sensitive public data while learning differentially private models, mostly for discriminative tasks (Papernot et al., 2017; 2018; Zhu et al., 2020), but also for generative modeling. Zhang et al. (2018) uses public data to pretrain discriminators for private GAN learning. Xu et al. (2019) uses public data to calibrate per parameter clipping bounds.

Recently, Triastcyn & Faltings (2020a;b) introduced Bayesian differential privacy for ML, a data-centric notion of differential privacy. It takes into account that a dataset's "typical" data is easier to protect than outliers. However, this approach has not been widely adopted by the community yet and the focus of our work is to introduce DP-Sinkhorn in the regular differential privacy setting.

## 6 CONCLUSIONS

We propose DP-Sinkhorn, a novel optimal transport-based differentially private generative modeling method. Our approach minimizes the empirical Sinkhorn loss in a differentially private manner and does in general not require any adversarial techniques that are challenging to optimize. Therefore, DP-Sinkhorn is easy to train and deploy, which we hope will help its adoption in practice. We also use a novel trick to force the optimal transport plan to couple same-label data together, thereby allowing for conditional data generation in the Sinkhorn divergence generative modeling framework. We experimentally validate our proposed method and demonstrate superior performance compared to the previous state-of-the-art on standard image classification benchmarks using data generated under DP. We also show differentially private synthesis of informative RGB images without using additional public data. Note that our main experiments only used a simple pixel-wise L2-loss as cost function. This suggests that in the DP setting, complexity in model and objective are not necessarily beneficial. We conclude that simple and robust models such as ours are a promising direction for differentially private generative modeling.

Future work includes improving DP-Sinkhorn's performance on RGB images and scaling it up to higher resolutions, extending it to other data types, using tailored kernel-based cost functions and incorporating auxiliary public data during training.

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

# A  OPTIMAL TRANSPORT VIA THE SINKHORN DIVERGENCE

In addition to the notations defined in Sec. 2.1, we denote the Dirac delta distribution at $x \in \mathcal{X}$ as $\delta_x$, and the standard $n$-simplex as $\mathcal{S}^n$.

Recall from Sec. 2.2 that, given a positive cost function $c : \mathcal{X} \times \mathcal{X} \mapsto \mathbb{R}^+$ and $\epsilon \geq 0$, the Entropy Regularized Wasserstein Distance is defined as:

$$W_{c,\epsilon}(\mu,\nu) = \min_{\pi \in \Pi} \int c(x,y)\pi(x,y) + \epsilon \int \log \frac{\pi(x,y)}{d\mu(x)d\nu(y)} d\pi(x,y) \tag{7}$$

where $\Pi = \left\{ \pi(x,y) \in \mathcal{P}(\mathcal{X} \times \mathcal{X}) | \int \pi(x,\cdot)dx = \nu, \int \pi(\cdot,y)dy = \mu \right\}$.

We use the Sinkhorn divergence, as defined in Feydy et al. (2019).

**Definition A.1** *(Sinkhorn Loss) The Sinkhorn loss between measures $\mu$ and $\nu$ is defined as:*

$$S_{c,\epsilon}(\mu,\nu) = 2W_{c,\epsilon}(\mu,\nu) - W_{c,\epsilon}(\mu,\mu) - W_{c,\epsilon}(\nu,\nu) \tag{8}$$

For modeling data-defined distributions, as in our situation, an empirical version can be defined, too. Note that we use a slightly different notation from the main text because it is more convenient to deal with empirical distributions rather than samples when relating to the dual formulation later on.

**Definition A.2** *(Empirical Sinkhorn loss) The empirical Sinkhorn loss computed over a batch of $N$ generated examples and $M$ real examples is defined as:*

$$\hat{S}_{c,\epsilon}(\hat{\mu},\hat{\nu}) = 2C_{XY} \odot P^*_{\epsilon,X,Y} - C_{XX} \odot P^*_{\epsilon,X,X} - C_{YY} \odot P^*_{\epsilon,Y,Y} \tag{9}$$

*where $\hat{\mu} = \frac{1}{N}\sum_{i=1}^{N} \delta_{x_i}$, and $\hat{\nu} = \frac{1}{M}\sum_{j=1}^{M} \delta_{y_j}$. For two samples $A \in \mathcal{X}^N$ and $B \in \mathcal{X}^M$, $C_{AB}$ is the cost matrix between $A$ and $B$, and $P^*_{\epsilon,A,B}$ is an approximate optimal transport plan that minimizes Eqn. 1 computed over $A$ and $B$.*

$P^*_\epsilon$ is arrived at by iterating the dual potentials.

Cuturi (2013) and Feydy et al. (2019) have shown the following dual formulation for the discritized version of $\hat{W}_{c,\epsilon}$:

$$\hat{W}_{c,\epsilon}(\hat{\mu},\hat{\nu}) = \max_{f,g \in \mathcal{S}^N \times \mathcal{S}^M} \langle \hat{\mu}, f \rangle + \langle \hat{\nu}, g \rangle - \epsilon \langle \hat{\mu} \otimes \hat{\nu}, \exp(\frac{1}{\epsilon}(f \oplus g - C)) - 1 \rangle, \tag{10}$$

where $\otimes$ denotes the product measure and $\oplus$ denotes the "outer sum" such that the output is a matrix of the sums of pairs of elements from each vector. Then, the optimal transport plan $P^*_\epsilon$ relates to the dual potentials by $P^*_\epsilon = \exp(\frac{1}{\epsilon}(f \oplus g - C))(\hat{\mu} \otimes \hat{\nu})$. Thus, once we find the optimal $f$ and $g$, we can obtain $P^*_\epsilon$ through this primal-dual relationship. We also know the first-order optimal conditions for $f$ and $g$ through the Karush Kuhn Tucker theorem:

$$f_i = -\epsilon \log \sum_{j=1}^{M} \exp(\log(\hat{\nu}_j) + \frac{1}{\epsilon}g_j - \frac{1}{\epsilon}C(x_i,y_j)) \quad g_j = -\epsilon \log \sum_{i=1}^{N} \exp(\log(\hat{\mu}_i) + \frac{1}{\epsilon}f_i - \frac{1}{\epsilon}C(x_i,y_j))$$
$$\tag{11}$$

To optimize $f$ and $g$, it suffices to apply the Sinkhorn algorithm (Cuturi, 2013), see Algorithm 3 in the main text. Readers can refer to Feydy (2020) for further details.

# B  DIFFERENTIAL PRIVACY

As discussed in Sec. 2.3, differential privacy is the current gold standard for measuring the privacy risk of data releasing programs. It is defined as follows (Dwork et al., 2006):

**Definition B.1** *(Differential Privacy) A randomized mechanism $\mathcal{M} : \mathcal{D} \rightarrow \mathcal{R}$ with domain $\mathcal{D}$ and range $\mathcal{R}$ satisfies $(\varepsilon,\delta)$-DP if for any two adjacent inputs $d, d' \in \mathcal{D}$ differing by at most one entry, and for any subset of outputs $S \subseteq \mathcal{R}$ it holds that*

$$\mathbf{Pr}\left[\mathcal{M}(d) \in S\right] \leq e^\varepsilon \mathbf{Pr}\left[\mathcal{M}(d') \in S\right] + \delta. \tag{12}$$

**Gradient perturbation**: For a parametric function $f_\theta(x)$ parameterized by $\theta$ and loss function $L(f_\theta(x), y)$, usual mini-batched first-order optimizers update $\theta$ using gradients $\mathbf{g}_t = \frac{1}{N} \sum_{i=1}^{N} \nabla_\theta L(f_\theta(x_i), y_i)$. Under gradient perturbation, the gradient $\mathbf{g}_t$ is first clipped in $L_2$ norm by constant $\Delta$, and then noise sampled from $\mathcal{N}(0, \sigma^2 \mathbb{I})$ is added. Since differential privacy is closed under *post-processing*—releasing any transformation of the output of an $(\varepsilon, \delta)$-DP mechanism is still $(\varepsilon, \delta)$-DP (Dwork & Roth, 2014)—the parameters $\theta$ are also differentially private. The relation between $(\varepsilon, \delta)$ and the perturbation parameters $\Delta$ and $\sigma$ is provided by the following theorem:

**Theorem B.1** *For $c^2 > 2\log(1.25/\delta)$, Gaussian mechanism with $\sigma \geq c\Delta/\varepsilon$ satisfies $(\varepsilon, \delta)$ differential privacy. (Dwork & Roth, 2014)*

**Subsampling**: In stochastic gradient descent (SGD) and related methods, randomly drawn batches of data are used in each training step instead of the full dataset. This subsampling of the dataset can provide amplification of privacy protection since the privacy of any record that is not in the batch is automatically protected. Privacy bounds for various subsampling methods have been extensively studied and applied (Dwork et al., 2006; Wang et al., 2019; Balle et al., 2018; Zhu & Wang, 2019).

**Composition**: SGD requires the computation of the gradient to be repeated every iteration. The repeated application of privacy mechanisms on the same dataset is analyzed through *composition*. Composition of the Gaussian mechanism has been first analyzed by Abadi et al. (2016) through the moments accountant method.

We utilize the often used Rényi Differential Privacy (Mironov, 2017) (RDP), which is defined through the Rényi divergence between mechanism outputs on adjacent datasets:

**Definition B.2** *(Rényi Differential Privacy) A randomized mechanism $\mathcal{M} : \mathcal{D} \to \mathcal{R}$ with domain $\mathcal{D}$ and range $\mathcal{R}$ satisfies $(\alpha, \varepsilon)$-RDP if for any adjacent $d, d' \in \mathcal{D}$ it holds that*

$$D_\alpha(\mathcal{M}(d) | \mathcal{M}(d')) \leq \varepsilon, \tag{13}$$

*where $D_\alpha$ is the Rényi divergence of order $\alpha$. Also, any $\mathcal{M}$ that satisfies $(\alpha, \varepsilon)$-RDP also satisfies $(\varepsilon + \frac{\log 1/\delta}{\alpha - 1}, \delta)$-DP.*

As discussed in the main text, RDP is a well-studied formulation of privacy that allows tight composition of multiple queries—training iterations in our case—and can be easily converted to standard definitions of DP with definition B.2. Recall that for clipping threshold $\Delta$ and standard deviation of Gaussian noise $\sigma$, the Gaussian mechanism satisfies $(\alpha, \alpha\Delta^2/(2\sigma^2))$-RDP (Mironov, 2017). Privacy analysis of a gradient-based learning algorithm entails accounting for the privacy cost of single queries, which corresponds to training iterations in our case, possibly with subsampling due to mini-batched training. The total privacy cost is obtained by summing up the privacy cost across all queries or training steps, and then choosing the best $\alpha$.

For completeness, the Rényi divergence is defined as: $D_\alpha(P|Q) = \frac{1}{\alpha} \log \mathbb{E}_{x \in Q} \left[ \frac{P(x)}{Q(x)} \right]^\alpha$.

## C  ALGORITHMS

---

**Algorithm 2** Poisson Sample

**Input** : $d = \{(\mathrm{y}, \mathrm{l}) \in \mathcal{X} \times \{0, ..., L\}\}^M$, sampling ratio $q$
**Output**: $\mathbf{Y} = \{(\mathrm{y}_j, \mathrm{l}_j) \in \mathcal{X} \times \{0, ..., L\}\}_{j=1}^m$, $m \geq 0$
$\mathrm{s} = \{\sigma_i\}_{i=1}^M \overset{i.i.d.}{\sim} \mathrm{Bernoulli}(q)$
$\mathbf{Y} = \{d_j | \mathrm{s}_j = 1\}_{j=1}^m$

---

**Algorithm 3** Sinkhorn Algorithm $\hat{W}_\epsilon(\mathbf{X}, \mathbf{Y})$

**Input:** $\mathbf{X} = \{\mathrm{x}\}^n, \mathbf{Y} = \{\mathrm{y}\}^m, \epsilon$
**Output:** $W_\epsilon$
$\forall(i,j), C_{[i,j]} = c(\mathbf{X}_i, \mathbf{Y}_j)$
$\mathbf{f}, \mathbf{g} \leftarrow \vec{0}$
$\hat{\mu}, \hat{\nu} \leftarrow \mathrm{Unif}(n), \mathrm{Unif}(m)$
**while** not converged **do**
 $\forall i, \mathbf{f}_i \leftarrow -\epsilon \log \sum_{k=1}^m \exp(\log(\hat{\nu}_k) + \frac{1}{\epsilon}\mathbf{g}_k - \frac{1}{\epsilon}C_{[i,k]})$
 $\forall j, \mathbf{g}_j \leftarrow -\epsilon \log \sum_{k=1}^n \exp(\log(\hat{\mu}_k) + \frac{1}{\epsilon}\mathbf{f}_k - \frac{1}{\epsilon}C_{[k,j]})$
**end while**
$W_\epsilon = \langle \hat{\mu}, \mathbf{f} \rangle + \langle \hat{\nu}, \mathbf{g} \rangle$

---

# D  EXPERIMENT DETAILS

## D.1  DATASETS

MNIST and Fashion-MNIST both consist of 28x28 grayscale images, partitioned into 60k training images and 10k test images. The 10 labels of the original classification task correspond to digit/object class. For calculating FID scores, we repeat the channel dimension 3 times. CelebA is composed of ∼200k colour images of celebrity faces tagged with 40 binary attributes. We downsample all images to 32x32, and use all 162770 train images for training and all 19962 test images for evaluation. Generation is conditioned on the gender attribute.

## D.2  CLASSIFIERS

For logistic regression, we use scikit-learn's implementation, using the L-BFGS solver and capping the maximum number of iterations at 5000. The MLP and CNN are implemented in PyTorch. The MLP has one hidden layer with 100 units and a ReLU activation. The CNN has two hidden layers with 32 and 64 filters, and uses ReLU activations. We train the CNN with dropout ($p = 0.5$) between all intermediate layers. Both the MLP and CNN are trained with Adam under default parameters, and use 10% of training data as holdout for early stopping. Training stops after no improvement is seen in holdout accuracy for 10 consecutive epochs.

## D.3  ARCHITECTURE, HYPERPARAMETER, AND IMPLEMENTATION

Our DCGAN-based architecture uses 4 transposed convolutional layers with ReLU activations at the hidden layers and tanh activation at the output layer. A latent dimension of 12 and class embedding dimension of 4 is used for MNIST and Fashion-MNIST experiments. CelebA experiments use a latent dimension of 32 and embedding dimension of 4. The latent and class embeddings are concatenated and then fed to the convolutional stack. The first transposed convolutional layer projects the input to $256 \times 7 \times 7$, with no padding. Layers 2,3 and 4 have output depth $[128, 64, 1]$, kernel size $[4, 4, 3]$, stride $[2, 2, 1]$, and padding $[1, 1, 1]$.

Our BigGAN-based architecture uses 4 residual blocks of depth 256, and a latent dimension of 32. Each residual block consists of three convolutional layers with ReLU activations and spectral normalization between each layer. Please refer to Brock et al. (2019) for more implementation details. Our implementation is based on `https://github.com/ajbrock/BigGAN-PyTorch`.

In our experiments with the adversarially trained feature extractor, we used a simple feature extractor with 3 convolutional layers with hidden depth of 64 and output depth of 32. Between each convolutional layer are batchnorm and ReLU layers, followed a $2 \times 2$ maxpool layer. The first two layers have kernel size 3 and padding 1, while the last layers have kernel size 1 with no padding. For the generator $G_\theta$ and feature extractor $\phi_\omega$, the adversarial loss objective can be formally expressed as:

$$\min_\theta \max_\omega \hat{S}_{c(\omega),\epsilon}(\hat{\mu}(\theta), \hat{\nu}) = \min_\theta \max_\omega 2\langle c_\phi(G_\theta(z), \mathrm{y}), P_\epsilon^*(G_\theta(z), \mathrm{y})\rangle$$
$$- \langle c_\phi(G_\theta(z), G_\theta(z)), P_\epsilon^*(G_\theta(z), G_\theta(z))\rangle$$
$$- \langle c_\phi(\mathrm{y}, \mathrm{y}), P_\epsilon^*(\mathrm{y}, \mathrm{y})\rangle$$

Where $c_\phi(\mathrm{a}, \mathrm{a}) = ||[\mathrm{a}_{img}, \alpha_c \mathrm{a}_{label}, \alpha_a \frac{\phi(\mathrm{a}_{img})}{||\phi(\mathrm{a}_{img})||_2}]^T - [\mathrm{b}_{img}, \alpha_c \mathrm{b}_{label}, \alpha_a \frac{\phi(\mathrm{b}_{img})}{||\phi(\mathrm{b}_{img})||_2}]^T||_2^2 = c_{adv}([\mathrm{a}_{img}, \mathrm{a}_{label}], [\mathrm{b}_{img}, \mathrm{b}_{label}])$

Hyperparameters of the Sinkhorn loss used were: $\alpha_c = 15$, and entropy regularization $\epsilon = 0.05$ in MNIST and Fashion-MNIST experiments. $\epsilon = 5$ is used for CelebA experiments. We use the implementation publically available at `https://www.kernel-operations.io/geomloss/api/install` and all other hyperparameters are kept at their default values. For all experiments, we use the Adam (Kingma & Ba, 2015) optimizer with learning rate $10^{-5}$, $\beta = (0.9, 0.999)$, weight decay $2 \times 10^{-5}$.

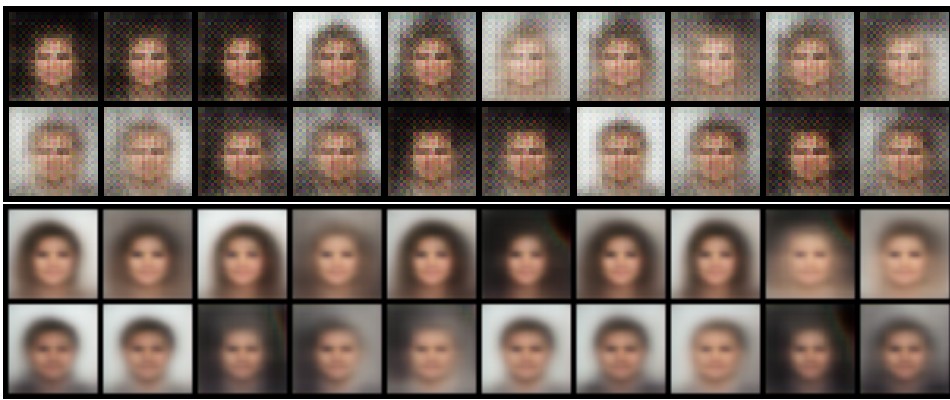

Figure 4: Additional DP-Sinkhorn generated images under $(10, 10^{-6})$differential privacy. Top two rows use DCGAN based generator, while bottom two rows use BigGAN based generator.

### D.4 IMPLEMENTATION OF DIFFERENTIAL PRIVACY

For privacy accounting, we use the implementation of the RDP Accountant available in Tensorflow Privacy.[2] All experiments employing the pixel-wise $L2$ cost use Poisson sampling, and are amenable to the analysis implemented in `compute_rdp`. For the experiments with the batch-wise feature extractor, examples are batched into records at the start of training, and a single record is drawn at random in each iteration. The privacy amplification for this fixed-size sampling scheme is studied in Wang et al. (2019), and we use the author's implementation of Theorem 9 for bounding the RDP cost of our queries.

For MNIST and Fashion-MNIST results reported in the main body, we use a noise scale of $\sigma = 1.1$ and a batch size of 50 resulting in $q = 1/1200$, which gives us $\sim 3.4$ million training iterations to reach $\varepsilon = 10$ for $\delta = 10^{-5}$. For the non-private runs, we use a batch size of 500, which improves image quality and diversity. When training with DP, increasing batch size significantly increases the privacy cost per iteration, resulting in poor image quality for fixed $\varepsilon = 10$. Image gradient perturbation used clipping norm bound of 0.5, while the parameter gradient perturbation variant used clipping norm bound of 1.

For CelebA results reported in the main body, we use a noise scale of $\sigma = 0.8$ and a batch size of 200 resulting in $q = 0.00123$. At $\delta = 10^{-6}$, we train for 1.1 million steps to reach $\varepsilon = 10$.

## E ADDITIONAL RESULTS

We evaluate the impact of architecture choice on the performance in the CelebA task by comparing DP-Sinkhorn+BigGAN with DP-Sinkhorn+DCGAN, under pixel loss. Results are summarized in Table 3 and visualized in Figure 4. Qualitatively, despite reaching lower FID score, the DCGAN-based generator's images have visible artifacts that are not present in models trained with BigGAN-generators.

Table 3: Differetially private image generation results on downsampled CelebA.

| Method | DP-$\epsilon$ | FID | Acc (%) | |
| --- | --- | --- | --- | --- |
| | | | MLP | CNN |
| Real data | $\infty$ | 1.1 | 91.9 | 95.0 |
| DCGAN+DP-Sinkhorn | 10 | 156.7 | 74.96 | 74.62 |
| BigGAN+DP-Sinkhorn | 10 | 168.4 | 76.18 | 75.79 |

---

[2]https://github.com/tensorflow/privacy/

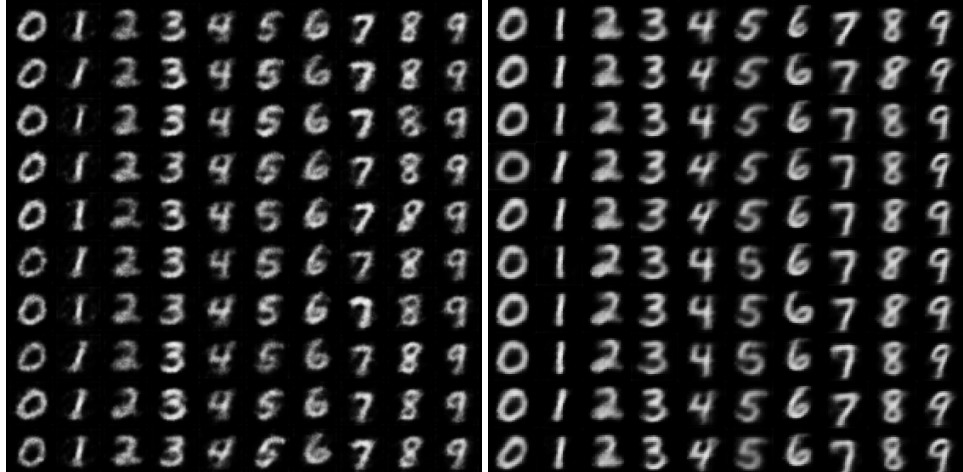

Figure 5: Additional images generated by DP-Sinkhorn, trained on MNIST. Left: Images generated using parameter gradient perturbation; these correspond to the "DP-Sinkhorn ($\nabla_\theta \hat{S}$)" row in the main table. Right: Images generated using gradient perturbation on generated images; these correspond to the "DP-Sinkhorn" row in the main results table.

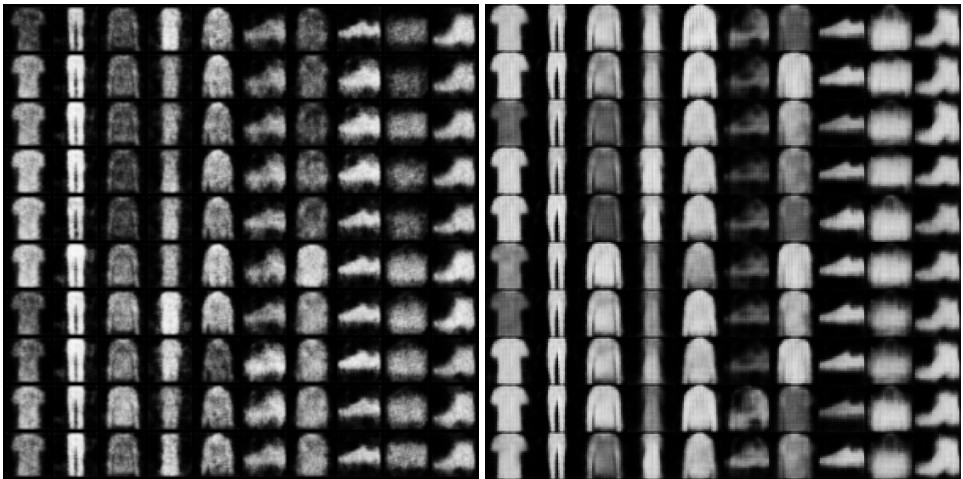

Figure 6: Additional images generated by DP-Sinkhorn, trained on Fashion-MNIST. Left: Images generated using parameter gradient perturbation; these correspond to the "DP-Sinkhorn ($\nabla_\theta \hat{S}$)" row in the main table. Right: Images generated using gradient perturbation on generated images; these correspond to the "DP-Sinkhorn" row in the main results table.

Table 4: Comparison of training DP-Sinkhorn with various optimizers and learning rates on MNIST. Batch size is 50 and DP $\epsilon = 10$. The first row corresponds to the configuration used in results reported in Table 1. Runs which did not converge are highlighted in red.

| Optimizer | Learning rate | FID | Acc (%) | | |
|---|---|---|---|---|---|
| | | | Log Reg | MLP | CNN |
| Adam | 1e-5 | 124.3 | 82.0 | 80.8 | 79.9 |
| SGD | 1e-5 | 135.0 | 77.1 | 77.6 | 78.8 |
| RMSProp | 1e-5 | 135.4 | 78.8 | 79.6 | 77.6 |
| Adam | 1e-6 | 137.7 | 78.4 | 79.5 | 78.6 |
| Adam | 1e-4 | 158.5 | 78.1 | 79.4 | 76.5 |
| Adam | 1e-3 | 158.2 | 76.8 | 77.3 | 76.1 |
| Adam | 1e-2 | 491 | 08.4 | 13.2 | 10 |
| SGD | 1e-6 | 137.7 | 78.4 | 79.5 | 78.6 |
| SGD | 1e-4 | 175.6 | 75.9 | 76.1 | 74.6 |
| SGD | 1e-3 | 360 | 10.2 | 5.6 | 8.7 |

Table 5: Comparison of training DP-Sinkhorn with various batch sizes and noise scales ($\sigma$) on MNIST. We use an Adam optimizer with learning rate 1e-5, and DP $\epsilon = 10$ is used as target to determine the number of epochs. The first row corresponds to the configuration used in results reported in Table 1.

| Batch size | Noise scale | FID | Acc (%) | | |
|---|---|---|---|---|---|
| | | | Log Reg | MLP | CNN |
| 50 | 1.1 | 124.3 | 82.0 | 80.8 | 79.9 |
| 50 | 0.9 | 135.6 | 80.3 | 78.5 | 77.0 |
| 50 | 1.3 | 140.6 | 82.1 | 80.5 | 79.6 |
| 100 | 0.9 | 139.1 | 78.8 | 78.8 | 79.2 |
| 100 | 1.1 | 131.4 | 79.4 | 79.8 | 77.6 |
| 100 | 1.3 | 122.1 | 78.9 | 78.6 | 76.7 |
| 200 | 2 | 113.8 | 73.1 | 73.4 | 72.8 |

