# OpenReview forum: "Differentially Private Generative Models Through Optimal Transport"
_ICLR.cc/2021/Conference — Reject_

### Official Review · AnonReviewer4 · 2020-10-28
**Proposed method is not well motivated and is not novel**

**Rating:** 4
**Confidence:** 3

**Review:**

This paper presents a differentially private method for training a generative model. The proposed method takes advantage of the Sinkhorn divergence to achieve robustness against the hyperparameters' choices. The authors also introduce a cost function enabling the generative model to generate images associated with a specific class label. The experimental results show that the proposed method outperforms the existing methods for learning a generative model in a differentially private manner. Furthermore, such a high accuracy can be achieved without the use of publicity available data.

The strong points of this paper are as follows:
-  The authors bring the Sinkhorn divergence-based learning of a generative model into the differentially private generative model learning problem.

The weak points of this paper are as follows:
- The proposed method is not clearly explained and thus is suspicious in the guarantee of differential privacy.
- The presented method is a direct application of Wang et al.'s moment account technique with Zhu et al.'s Poisson sampling. The originality is considerably low.
- This paper has no theoretical and experimental analysis about robustness against a hyperparameter choice, while the authors claim it as a contribution.
- The proposed approach is not well motivated. We can use some differentially private classification algorithm if the objective is high accuracy in downstream classification.

I recommend rejection of this paper because the proposed algorithm has low originality and is not well motivated. Also, the unclarity of the privacy guarantee is problematic.

The authors do not give a clear explanation of the proposed method. In particular, it is unclear if the proposed algorithm guarantees differential privacy. I guess the authors employ either the composition theorem or moment account technique to prove the algorithm's differential privacy; however, there is no privacy proof of the proposed algorithm. I could not confirm that the proposed method ensures differential privacy.

The proposed method is a straightforward application of the techniques from Wang et al. and Zhu et al. Also, defining the cost function as in Eq. 4 is a straightforward way to combine the multidimensional real-valued data and discrete label. I cannot find any original idea, except introducing the Sinkhorn divergence, in the proposed method.

The authors claim that the proposed method is robust against the choice of its hyperparameters. However, there is no evidence to support the claim. A theoretical or experimental analysis of robustness is necessary to claim it.

Why don't we employ the differentially private classification algorithm, such as M. Adabi et al. Deep Learning with Differential Privacy. In CCS'16. When the objective is high accuracy in the downstream classification, we can utilize such an algorithm directly. What is the benefit of employing the generative model-based privacy preservation? The differentially private classification algorithm can achieve high classification accuracy; for example, in Adabi et al.'s paper, the classification accuracy for MNIST with eps=10 is 97%; this value is significantly higher than that of the proposed method.


### Minor comments

- What is the definition of $\hat{S}$?

---

> ### Author Response · Authors · 2020-11-20
> **Thank you for the constructive feedback**
>
> We thank the reviewer for the constructive feedback. Below is our response to some specific concerns raised in your review.
> * The guarantee of differential privacy
>
> Regarding the guarantee of privacy, we used poisson subsampling with moments accountant for composing Renyi Differential Privacy (RDP) over training iterations, as proposed in Zhu et al. 2019 [1]. We used the implementation of this mechanism in Tensorflow-DP, with further details provided in appendix D.4. As this moments accountant technique is well established and our implementation by working with the Tensorflow-DP library is fairly standard, we are certain that our method is differentially private in both theory and practice. We will release our source code once the paper is published. We added a proof of our method’s privacy at the end of section 3. We chose to not include this in the original manuscript, since the technique is a standard application of privacy results for the Gaussian mechanism in RDP. The goal of our paper is indeed not to introduce a novel privacy mechanism, but rather to introduce optimal transport to the field of differential privacy as a promising method for learning differentially private generative models (see below for motivation for that).
>
> * Originality of the proposed method
>
> Regarding originality, our work highlights that we can learn private generative models that are more useful than recent state-of-art methods (GS-WGAN) using an easy-to-train, non-adversarial loss formulation (Sinkhorn Divergence) and standard differential privacy tools. Our motivation is that as a non-adversarial optimal transport loss, Sinkhorn Divergence converges more stably than GANs, which makes it naturally suited for adapting to differential privacy. We are particularly interested in the usefulness of the synthesized data for downstream tasks, for which we choose classification of MNIST and FashionMNIST. This has emerged as the standard benchmark for differentially private generative models. A strong classifier that generalizes to real test data can only be trained if the data used for training is diverse. GANs, which are used by most competitive works, tend to suffer from instabilities during training due to their adversarial training scheme. These instabilities can manifest as mode dropping and imbalanced covering of the data distribution, thereby synthesizing imbalanced data for training the classifier. This would result in a sub-optimal classifier. The advantage of our optimal transport-based method is precisely that it does not require any adversarial objectives in its standard form and does not suffer from any noticeable mode-dropping problems. We indeed found our images to be diverse, as seen by samples (see variance of shape and style of digits and grayscale color of outfit pieces in additional samples in Appendix E) and supported by the strong classification results. Our results are still worse in FID than the state-of-the-art due to slight blur. However, for downstream tasks diversity seems to trump sharpness. Generally, training instabilities and mode dropping problems in GANs may be exacerbated by the gradient noise applied during training for ensuring privacy. We think that it is unclear whether GANs will be as successful for training differentially private generative models as they are for non-private generative modeling. We believe that optimal transport provides a promising and very simple and robust framework for differentially private generative modeling, which we try to demonstrate with our results. We see our method’s simplicity as its strength.
>
> More generally, in non-private settings, a growing number of works have studied the application of optimal transport in learning deep generative models due to their desirable convergence properties, yet previous works on private generative learning have mostly focused on adversarial training methods. Our method fills the knowledge gap by bridging an important branch of generative models with the differentially private setting. Our work is conceptually most related to DP-MERF [2], which uses maximum mean discrepancy, another type of integral probability metric. Compared to DP-MERF, samples generated by our method are much more representative of the training data at a weaker privacy requirement of epsilon=10, whereas DP-MERF excels at the strong privacy regime of epsilon < 1, but fails to scale in utility with weaker privacy requirements.

---

> > ### Author Response · Authors · 2020-11-20
> > **Response to reviewer 4 continued**
> >
> > * Experimental analysis of robustness
> >
> > Regarding our claim about robustness, we have experimentally verified that our method converges stably on a wide range of learning rates, batch sizes and for three different optimizers (SGD, Rmsprop, Adam). We have added these results to the updated manuscript in Appendix E, Tables 4 and 5, and briefly discuss them at the end of Section 4.2. Our method also has fewer hyperparameters than GANs that involve separate discriminator learning rates, update schedules, and gradient penalty. Overall, DP-Sinkhorn can be trained with minimal hyperparameter optimization, aiding its prospect for adoptance by practitioners. Please also see section 2.2 of the updated manuscript for why learning with Sinkhorn divergence is naturally stabler than GANs.
> >
> > * Motivation for privately learning generative models
> >
> > Regarding why we choose to train generative models rather than discriminative models, we agree that if a discriminative model can be trained directly, then it will (likely always) perform better than first training a generator, and then training a discriminator on generated data, irrespective of whether privacy is required. However, there are practical considerations where this is not always feasible. Please see our general response that is addressed towards all reviewers regarding this point.
> > S_hat is the empirical Sinkhorn divergence computed on a given batch, we have updated our manuscript to define this in the main text (Section 3).
> >
> > We would like to thank the reviewer again for the helpful feedback. If there are any other remarks or questions, we would be happy to discuss them in this forum.
> >
> > [1] Yuqing Zhu, Yu-Xiang Wang. “Poisson Subsampled Renyi Differential Privacy” (2019)
> >
> > [2] Frederik Harder, Kamil Adamczewski, Mijung Park. “DP-MERF: Differentially Private Mean Embeddings with Random Features for Practical Privacy-Preserving Data Generation” (2020)

---

### Official Review · AnonReviewer3 · 2020-10-29
**Method not so novel, experiments not so convincing**

**Rating:** 4
**Confidence:** 3

**Review:**

The paper proposes a method for training OT GANs using differentially private sinkhorn algorithm. The idea is very simple - train GANs with sinkhorn divergences and add Gaussian noise to the gradient of output wrt generated samples. So, the novelty by itself is minimal as sinkhorn GANs have previously been proposed. The method merely adds Gaussian noise to gradients. The DP analysis is also not very different.

I am generally ok with incremental improvements in method if experimental results are strong. This doesn't seem to be the case in this paper. Two datasets are considered - MNIST and CelebA. In MNIST, we observe that the algorithm performs poorly compared to GS-WGAN both in inception score and FID. The method gets better accuracy though. This suggests that the method does not produce diverse samples. For example, consider a case of a mode collapse where a GAN generates only one sample per class. In this case, FID scores will be poor. However, if the generated sample correcly corresponds to the class, classification score will be high. My guess is similar thing is happening here. This is evident even in Fig 2 where the method has much poor sample diversity compared to GS-WGAN. Hence, the model itself is not that great.

In CelebA, despite using BigGAN style architecture, there is significant blur. The real challenge in differential private GANs is to generate samples with good quality while still satisfying privacy constraints.  This doesnt seem to be happening here. Also, comparison with GS-WGAN or other differentially private GANs are missing in CelebA experiment.

For conditional generation, is concatenation of data and labels in cost function a good strategy? It looks like both data and label space is very different, and I am wondering if such type of concatenation might be weak. Can you comment on this.

---

> ### Author Response · Authors · 2020-11-20
> **Thank you for the constructive comments**
>
> We thank the reviewer for the constructive feedback. Below is our response to some specific concerns raised in your review.
>
> * “Two datasets are considered… [DP-Sinkhorn] performs poorly compared to GS-WGAN both in inception score and FID”
>
> We would like to first highlight that we conducted experiments on three datasets: MNIST, Fashion-MNIST, and CelebA. MNIST and Fashion-MNIST are the de-facto benchmarks for differentially private generative learning in the image domain, as they have been used by many related works. The improvement in downstream classifier accuracy with DP-Sinkhorn is most pronounced on the Fashion-MNIST dataset, with an absolute improvement of 5.4% (8% relative) over GS-WGAN under respective best case scenarios. The metrics we evaluated on were classification accuracies and FID; we chose not to evaluate Inception score since we couldn’t determine how inception score was calculated in previous works when evaluating on Fashion-MNIST.
>
> * Regarding FID and sample diversity.
>
> Regarding FID, DP-Sinkhorn with L2 loss indeed produced images with higher FID than GS-WGAN, while comparing favorably against all other methods. We found that if we suppress pixels with brightness less than 0.5 in images generated by DP-Sinkhorn, we obtain images with worse visual quality. Clearly, this modification could not improve sample diversity either. Yet, the modified images score better FID (54.16) than GS-WGAN (61.3). This indicates that our method indeed produces diverse samples, but these images are slightly blurry and hence score poorly on FID which emphasizes texture. For fair comparison, we didn’t do any such tricks for the results presented in the paper, though. This also suggests that FID is not a reliable metric for this low resolution grayscale data and we deem the classification task more important. Importantly, we believe that DP-Sinkhorn performs better on classification despite the blur precisely because our images are more diverse and hence provide better generalization for downstream classifiers. The importance of diversity for downstream accuracy is even more pronounced in our newly added ablation experiments where we add noise on parameter gradients instead of image gradients. DP-Sinkhorn with parameter gradients produced even noisier images, yet the downstream classifier accuracy is still better than GS-WGAN. We hypothesize that this actually suggests that GS-WGAN and the other GAN-based approaches suffer from a certain amount of mode-dropping, while our approach does not. We attribute this to our robust optimal transport-based training approach that does avoid any potentially unstable adversarial objectives (in fact, we don’t consider our method a type of GAN, as it doesn’t train in an adversarial fashion. Rather, it directly optimizes the primal formulation of the optimal transport distance between the target and generated distribution via the Sinkhorn divergence - also see expanded explanation in section 2.2).
> We also added additional figures with more samples drawn from our model in Appendix E to demonstrate that it indeed produces diverse samples (see, for example, variations in shape and orientation of digits and grayscale color in outfit pieces).
>
> * Regarding the CelebA experiments
>
> We emphasize that our goal is to produce “useful” synthetic data for downstream tasks. As Anonreviewer 2 has also pointed out, it is impossible to perfectly learn a synthetic distribution (in the language of statistics, match a target distribution up to arbitrary order statistics) while also enforcing differential privacy.  DP-Sinkhorn shows that it is indeed possible to synthesize useful data when training private generators on RGB images. To our best knowledge, no previous works have attempted training differentially private generative models on RGB images without additional public data. We want to explore this important yet missing application with our proposed method. Although we find the generated data useful for downstream tasks, there is clearly still room for improvement. Future works might explore metrics (replacing pixel space L2) that algorithmically encode our priors about natural images. We couldn’t evaluate GS-WGAN on CelebA as code for GS-WGAN was not available at the time of submission. We are currently trying to train GS-WGAN on CelebA, but we expect experiments to run past the end of the discussion period.

---

> > ### Author Response · Authors · 2020-11-20
> > **Response to reviewer 3 continued**
> >
> > * Is concatenation of data and labels in cost function a good strategy?
> >
> > While simple, our class-conditioning strategy by concatenation is theoretically grounded as it equates to performing optimal transport in the joint space of images and labels. We elaborate more on this point in section 3 of the revised manuscript. In more detail, though, the key modelling decision is in fact not whether the label is concatenated with the image, but which cost function to use when computing the Sinkhorn loss. In our case, by concatenating the label to the image, we are also assuming a squared L2 norm for measuring similarly between labels. One could argue that squared L2 norm is an awkward choice for measuring similarity between one-hot vectors. However, we found that squared L2 was indeed sufficient for our experiments, as we have not observed any instances of the class-conditioning failure. Hence, we chose to stick to the simplest choice, in particular because we are using the L2 norm already for the image cost itself. That being that, we completely agree that more tailored cost functions may be needed for more complex label spaces (one could imagine, for example, a hierarchical softmax for hierarchical labels). We leave these explorations for future work.
> >
> > If there are any other remarks or questions, we would be happy to discuss them in this forum.

---

### Official Review · AnonReviewer2 · 2020-10-30
**new loss function for DP synethic image data**

**Rating:** 6
**Confidence:** 2

**Review:**

This paper proposes a novel architecture and training process for learning differentially private synthetic data generators.

Rather than taking an explicit adversarial GAN approach, the paper looks at minimizing a regularized Wasserstein distance (called the Sinkhorn loss) between the empirical distribution of the data and the synthetic distribution.

Their approach brings two benefits: first, the Sinkhorn loss is more straightforward to optimize than traditional GANs. The authors don’t really discuss why this is, but my understanding is that the “adversary” is contained in the dual formulation of Wasserstein distance (as the minimum over Lipschitz functions of the difference in expectation between two distributions). Considering such a Lipschitz adversary simplifies optimization.

Second, the paper uses an idea first published by Chen et al (NeurIPS 2020) but described as independent work here: Instead of measuring the gradient of the full parametric distribution of the generator, the authors measure the gradient “at the generated image level”. This gives a lower-dimensional gradient (requiring less noise).

Overall, the paper achieves a moderate improvement over the work of Chen et al with respect to a few simple measures of accuracy (namely, how well two DNN models do at classifying real data when they are trained on synthetic data). It does considerably worse than Chen et al’s method with respect to the FID score, a measure of visual similarity.

I generally like the idea of exploring alternate training strategies. Nevertheless, I have several reservations about the paper:

1. The relationship to Chen et al. It seems like much of the gain relative to previous work comes from the way gradients are compressed. But this idea appears already in the NeurIPS paper of Chen et al. It is unclear to me how to handle the relative priority of the papers.
2. Overly simplistic accuracy measures: Wasserstein (or Sinkhorn) distance is a natural loss function, but it is only likely to be a really good measure of accuracy when it is very small. The DP literature, since the pioneering work of Blum, Ligett, and Roth, has focused on using as a loss function the minimum over a (potentially enormous but) task-specific set of queries as the ultimate loss. We understand as a field that general-purpose synthetic data isn’t really possible (it preserves “too many statistics, too accurately” to avoid attacks). How can the framework here be adapted to such accuracy measures? How well does DP Sinkhorn do when measured against such task-specific accuracy? Can DP Sinkhorn be applied to anything except images (and if so, how well does it do)?

Overall, I find the paper interesting but perhaps borderline.

---

> ### Author Response · Authors · 2020-11-20
> **Thank you for the constructive comments**
>
> We thank the reviewer for the constructive feedback. Below is our response to some specific points raised in your review.
>
> * Why Sinkhorn divergence is easier to optimize than GANs.
>
> When compared to WGAN, learning with the Sinkhorn divergence has distinct differences. First, the Sinkhorn divergence is computed under the primal formulation of OT, whereas WGAN's loss is computed under the dual formulation. While both are approximations to the exact Wasserstein distance, the source of the approximation error differs. The Sinkhorn divergence uses entropic regularization to ensure linear convergence when finding the optimal transport plan $\pi$ [1]. Its two sources of error are the suboptimality of the transport plan and bias introduced by entropic regularization. With the guarantee of linear convergence, $\pi$ can converge to optimality by using enough iterations, thereby allowing control of the first error.
> The second source of error can be controlled by using small values of $\epsilon$, which we found to work well in practice. In contrast, WGAN's source of error lies in the sub-optimality of the dual potential function. Since this potential function is parameterized by an adversarially trained deep neural network, it enjoys neither convergence guarantees nor feasibility guarantees.
> Furthermore, the adversarial training scheme can produce oscillatory behavior, where the discriminator and generator change abruptly every iteration to counter the strategy of the other player from the previous iteration [2]. These shortcomings contribute to WGAN's problems of non-convergence, which in turn can lead to mode dropping. In contrast, training with the Sinkhorn divergence does not involve any adversarial training at all, converges more stably, and reaps the benefits of OT metrics at covering modes. This stability is our key motivation for using the Sinkhorn approach for learning differentially private generative models, where stability in the adversarial GAN setting may be even harder to achieve due to the additional perturbations for guaranteeing differential privacy.
> We’ve expanded our discussion on differences between Sinkhorn divergence and GANs in section 2.2 of the manuscript.
>
> * Impact of gradient perturbation on generated images
>
> We have added an ablation study to our MNIST and FashionMNIST experiments in section 4.2. Our results indicate that gradient perturbation on generated images is still superior to doing so on the parameters, but DP-Sinkhorn with parameter gradient perturbation still slightly outperforms all baselines on the classification task. Qualitatively, the images generated by parameter gradient perturbation appear worse (see Appendix E), yet the downstream classification accuracy is close to the one achieved by image gradient perturbation. We think this is due to the fact that downstream classifiers are dependent on sample diversity to achieve generalization, and images generated by DP-Sinkhorn with parameter gradients are diverse enough to train (relatively) good classifiers despite having a noisier appearance. We hypothesize that the baselines, mostly being advarially trained GAN-based methods, suffer from some amount of mode dropping compared to our method. We attribute this to our robust optimal transport-based training approach (see previous bullet point).

---

> > ### Author Response · Authors · 2020-11-20
> > **Response to reviewer 2 continued**
> >
> > * Adapting to task specific measures
> >
> > We agree with the observation that differentially private training of generators that resemble the real distribution by every measure is impossible. Thus, it would make sense to use task-specific measures when training, such that the generator resembles the real distribution in features that are important for the downstream task. In fact, optimal transport is a very flexible framework and what the generator learns depends entirely on what the cost function penalizes. Hence, in principle we could influence the type of features learned by the generator by modifying the cost function. While our class conditioning method is a first step in this direction, more sophisticated methods could certainly be deployed under the same framework. For example, if a feature extractor trained for the classification task was available a priori, then we could use this extractor in the cost function (replacing the adversarial feature extractor presented in 4.3). This would cause the generator to match the real distribution not in pixels, but in features that are useful for the classification task. This feature extractor would have to be either trained on public data, or with differential privacy. While this is an exciting direction, we think this is complementary to our main goal of promoting optimal transport-based generative learning (for its stability and ease of training) under differentially private settings. Future work could definitely improve upon our current framework in this direction. Please also see our overall response regarding our motivation for why we chose to work on generative learning rather than training a classifier directly.
> >
> > * Can DP Sinkhorn be applied to anything except images
> >
> > Absolutely! Although the architecture of the generator network would need to be tweaked to fit the specific data type, private learning with Sinkhorn divergence is a general approach that can be applied to any data type for which a cost function can be defined. For example, cross entropy can be used for categorical random variables, and L1 distance can be used for real vectors that are sparse. Optimal transport has been used as a metric on other complex data types such as language and graphs [3-5].
> > We chose not to perform evaluations on other data types, since recent related works have been mostly focused on image generation and we wanted to benchmark our framework against these works. However, this is indeed a very interesting direction for future research and arguably another strength of our OT-based approach. As described, adapting our method to other data types just means replacing the cost function and is therefore easy, whereas training GANs (the baselines are mostly based on GANs) on other data types can be very challenging, due to the adversarial training, which we avoid.
> >
> > If there are any other remarks or questions, we would be happy to discuss them in this forum.
> >
> > [1] Marco Cuturi. “Sinkhorn Distances: Lightspeed Computation of Optimal Transport” (2013)
> >
> > [2] Lars Mescheder, Sebastian Nowozin, Andreas Geiger. “The Numerics of GANs” (2017)
> >
> > [3] Matt J. Kusner, Yu Sun, Nicholas I. Kolkin, Kilian Q. Weinberger. “From Word Embeddings To Document Distances” (2015)
> >
> > [4] David Alvarez-Melis, Tommi S. Jaakkola, Stefanie Jegelka. “Structured Optimal Transport“ (2018）
> >
> > [5] Titouan Vayer, Laetitia Chapel, Remi Flamary, Romain Tavenard, Nicolas Courty. “Optimal Transport for structured data with application on graphs” (2019)

---

### Author Response · Authors · 2020-11-20
**Overall response and summary of manuscript update**

We thank all reviewers for the constructive feedback. We reply to some points that have been discussed by multiple reviewers here.

We agree that if we were only interested in training a differentially private classifier, then using a generative model as we do, may not be necessary. However, we believe that generative models have significant potential as a data sharing medium with differential privacy. We have expanded on our motivation for learning generative models in a differentially private manner in the introduction of the manuscript, and provide a more detailed explanation here. While gradient perturbation methods like DPSGD can be applied generally to training machine learning models, they have practical requirements that restrict their application. For instance, in a common setting where a data analyst (party wanting to use the data, such as a marketing company, medical ML start-up, etc.) wants to train a differentially private model on personal data held by a data curator (party in charge of safekeeping the database, such as government agencies, hospitals, mobile device operators, etc.), the data analyst would either: A. send their machine learning model and code to the curator and have the curator run DPSGD locally, or B. establish a high speed connection between the two parties, send model weights to the curator, and receive gradients back from the curator for updating the model. In either case, the analyst would have to disclose their model architecture to the curator, and the curator needs to perform computations locally that could become expansive if they are sharing data with multiple analysts. While the curator could theoretically use local differential privacy to treat the data before sending it to the analyst, the amount of noise required for adequate privacy protection will likely render the data useless. Our goal is to use privately learned generative models as a data sharing medium, where the curator can train a generator once, and share it with any number of analysts. The analysts could then use the generative model to produce synthetic data for training downstream tasks. This way, the computational burden on the curator is reduced in the long run, and the analyst would not need to transfer technology (that is likely vital for competitiveness) to the curator. Hence, sharing data by sharing a generative model can be considered a very general and versatile form of data sharing. The data analyst can use the data synthesized by the differentially private generative model in any way they like. Training a classifier, as we do in our paper, is merely one example. In principle, this data could be used to train other models, too. Hence, we argue that data sharing by generative modeling is a promising research direction in the area of differential privacy, which is why this task has been explored by several previous works before, for example [1-5]. Training a classifier with generated data under differential privacy for MNIST as well as FashionMNIST classification has emerged as a standard benchmark. Therefore, we choose this benchmark in our work as well.

Including further points addressed in detail in the other replies, we have made the following modifications to our manuscript:
* Expanded Introduction on motivating private generative models,
* Expanded Section 2.2 on differences between Sinkhorn divergence and other methods,
* Further explained in Section 3 why concatenating the label and images works, including two interpretations,
* Added a privacy proof of our method following standard arguments in Section 3,
* Additional qualitative results (synthetic images) in Appendix E, Figures 5 and 6.
* Ablation results on perturbing image gradients versus perturbing parameter gradients (Section 4.2),
* Hyperparameter sweep demonstrating training stability of DP-Sinkhorn under multiple learning rates and optimizer choices (Section 4.2 and Appendix E, Tables 4 and 5).


We hope that we have been able to address all reviewers’ feedback and we welcome further discussion. Thank you!

[1] Liyang Xie, Kaixiang Lin, Shu Wang, Fei Wang, Jiayu Zhou. “Differentially Private Generative Adversarial Network” (2018)

[2] Yunhui Long, Suxin Lin, Zhuolin Yang, Carl A. Gunter, Bo Li. “Scalable Differentially Private Generative Student Model via PATE” (2019)

[3] Reihaneh Torkzadehmahani, Peter Kairouz, Benedict Paten “DP-CGAN: Differentially Private Synthetic Data and Label Generation” (2020)

[4] Frederik Harder, Kamil Adamczewski, Mijung Park “DP-MERF: Differentially Private Mean Embeddings with Random Features for Practical Privacy-Preserving Data Generation” (2020)

[5] Dingfan Chen, Tribhuvanesh Orekondy, Mario Fritz “GS-WGAN: A Gradient-Sanitized Approach for Learning Differentially Private Generators” (2020)

---

### Decision · Program_Chairs · 2021-01-07
**Final Decision**

**Decision:**

Reject

**Comment:**

The paper proposes a DP method for generative modelling based on optimal transport. The reviewers agree that the novelty is limited in relation to prior work, while the results are not especially compelling either. So, even though this is a valid approach, correctness is not sufficient for acceptance at ICLR.